# The Significance of Calcium in Photosynthesis

**DOI:** 10.3390/ijms20061353

**Published:** 2019-03-18

**Authors:** Quan Wang, Sha Yang, Shubo Wan, Xinguo Li

**Affiliations:** 1College of Life Sciences, Shandong Normal University, Jinan 250014, China; quanwang0120@163.com; 2Biotechnology Research Center, Shandong Academy of Agricultural Sciences, Jinan 250100, China; yangsha0904@126.com; 3Scientific Observing and Experimental Station of Crop Cultivation in East China, Ministry of Agriculture, Jinan 250100, China; wanshubo2016@163.com; 4Shandong Academy of Agricultural Sciences and Key Laboratory of Crop Genetic Improvement and Ecological Physiology of Shandong Province, Jinan 250100, China

**Keywords:** calcium, chloroplasts, photosynthesis, calmodulin, energy dissipation

## Abstract

As a secondary messenger, calcium participates in various physiological and biochemical reactions in plants. Photosynthesis is the most extensive biosynthesis process on Earth. To date, researchers have found that some chloroplast proteins have Ca^2+^-binding sites, and the structure and function of some of these proteins have been discussed in detail. Although the roles of Ca^2+^ signal transduction related to photosynthesis have been discussed, the relationship between calcium and photosynthesis is seldom systematically summarized. In this review, we provide an overview of current knowledge of calcium’s role in photosynthesis.

## 1. Introduction

Calcium is involved in many pathways in plant cells, including plant growth and development, resistance to environmental stress [1,2], hormonal response [3], interaction with pathogenic microorganisms [4], and photosynthesis [5]. Calcium signals constitute a massive and complex signaling network in plant cells. The downstream signaling molecules activated by increasing levels of free calcium in these pathways are similar, such as calmodulin (CaM), reactive oxygen species (ROS), and respiratory burst oxidase homologues (Rbohs) [6]. Therefore, the responses of multiple pathways might be affected by only one calcium pathway with different response levels. Calcium channels exist in cytoplasmic membranes, cell nuclear membranes, and various organelle membranes [4]. Calcium can regulate the transcription and translation of genes that encode chloroplast proteins and enzymes, which are involved in the reactions of photosynthesis. 

Photosynthesis is the most extensive biosynthesis process on Earth, and it occurs in chloroplasts, which have a calcium pool [7]. When plants face environmental stimulus, calcium is released immediately from the calcium pool, triggering downstream events. Ca^2+^-binding proteins have been consistently found to reside in chloroplasts, but some are located in the chloroplast membrane, such as s-adenosylmethionine transporter-like (SAMTL) [8] and chloroplast inner envelope protein (TIC) [9]. The chloroplast membrane proteins can directly interact with the cytoplasm signal molecules; the chloroplast proteins regulated by calcium affect cytoplasm signaling pathways, and vice versa. Simultaneously, calcium in chloroplasts can regulate the photosynthetic pathway, which is the main source of energy supply for plant cells. When exposed to external stimulus, calcium oscillations trigger the downstream signals to protect plant cells. For example, the Ca^2+^ signal transduction pathway can regulate xanthophyll cycle-dependent non-photochemical quenching (NPQ) [10]. Whether this phenomenon suggests a connection between calcium and photosynthetic energy remains unknown. Photosynthesis is highly sensitive to environmental stress, and inappropriate environments will cause a decrease in photosynthesis. Stomatal and non-stomatal limitations are the two causes of inhibition of photosynthesis [11]. In this review, we attempt to elucidate the mechanisms of Ca^2+^-related pathways involved in photosynthesis (Figure 1). The photosynthesis-correlated proteins regulated by calcium are given in Table 1.

## 2. Mechanisms of Ca^2+^ Involved in Stomatal Movements

Stomata are important channels for plants to communicate with the environment, especially during photosynthesis. Each stoma comprises a pair of guard cells with a small amount of chloroplasts, and these chloroplasts are related to stomatal movement [46]. *Arabidopsis thaliana* mutants with no chloroplasts in guard cells show that the closure of stomata is greater than that in wild-type *A. thaliana* [47]. Chloroplasts are different between guard cells and mesophyll cells [48,49]. The chloroplasts in guard cells have many large starch grains, and their volume is larger than that in mesophyll cells [48]. Stomatal movement is regulated by the water content (e.g., soluble sugars) of guard cells, and osmotic substances also play an important role. The presence of amylase in guard cells can regulate water content through soluble sugars produced by the degradation of starch to regulate stomatal movement [48,50].

As a key signal of stomatal regulation [51], ROS is mainly produced in chloroplasts [52]. The production of hydrogen peroxide (H_2_O_2_) induced by abscisic acid (ABA) in guard cell chloroplasts is earlier than that in other locations [53]. The accumulation of H_2_O_2_ can inhibit amylase activity and reduce the sugar content [54]. Ca^2+^ sensor (CAS), which is located on the thylakoid membranes of a chloroplast, is a Ca^2+^-binding protein [12,13,14] associated with the production of H_2_O_2_ and nitric oxide (NO) in the stomatal closure pathway [55]. In CAS deletion mutations, the cytoplasm Ca^2+^ ([Ca^2+^]_cyt_) concentration decreases and stomatal closure is prevented [56]. However, both artificially induced [Ca^2+^]_cyt_ oscillation [56] and H_2_O_2_ treatment [57] can cause stomatal closure in mutants. Moreover, H_2_O_2_ affects stomatal movement by activating [Ca^2+^]_cyt_ channels [58], and CAS is involved in the generation of H_2_O_2_, which induces [Ca^2+^]_cyt_ oscillation by activating [Ca^2+^]_cyt_ channels and then causes stomatal closure. Some reports showed that [Ca^2+^]_cyt_ oscillations can also be induced by the inositol 1,4,5-trisphosphate (IP_3_) under external stress [59,60,61], and this phenomenon is related to CAS [61].

Additionally, NADPH oxidases RbohD and RbohF, which are also known as respiratory burst oxidase homologues, are both involved in stomatal movement [15,16]. Moreover, they have two Ca^2+^-binding EF-hand motifs [17]. RbohD/F [62] and phosphatidic acid (PA) produced by phospholipase D (PLD) [63] are related to ROS production, and the *PLDa1-*null *Arabidopsis* mutant [64] and rbohD/F depletion mutant [16] are insensitive to the pathway in which ABA induces stomatal closure. RbohD/F and PA are speculated to function in the same pathway. Further research showed that PA interacts with RbohD/F to increase NADPH activity, thereby affecting ROS production [64]. In addition, NO is involved in this pathway and plays a vital role [65]. When the activities of phospholipase C (PLC) and PLD are inhibited, NO-induced stomatal closure is also prevented [65]. Thus, NO acts upstream in the PLC and PLD pathway [65]. However, different studies showed that NO occurs in the downstream events of the pathway, in which PLD generates PA [64]. Stomatal movement is co-regulated by a signal network including the Ca^2+^ signal transduction pathway, plant hormone pathway, and ROS signal pathway, but the relationship between NO and PA on the stomatal movement pathway needs further research. Other signaling molecules involved in stomatal movement have been discussed in detail in another review [66].

## 3. Ca^2+^ Is Involved in the Processes of Photosynthetic Reaction

Photosystem 2 (PS2) is composed of membrane-related redox enzymes, and Ca^2+^ acts as a cofactor to participate in the formation of activation sites [21,67]. Oxygen-evolving complex (OEC), a component of PS2, is involved in the decomposition of water molecules [68]. Extrinsic PsbQ, PsbP, and PsbO are OEC proteins [19,22] that are closely related to CP47, α subunit of cytochrome b559 and a small subunit in PS2 [69]. PsbQ and PsbP require Cl^−^ and Ca^2+^ as essential co-factors [19,23]. PsbO-associated Ca^2+^ is from the Mn_4_CaO_5_ cluster [20,21], and PsbO is closely related to the stability of the Mn_4_CaO_5_ cluster [70]. Ca^2+^ also participates in the s-state cycle, which is associated with water decomposition [71]. Many studies have suggested that the Mn_4_CaO_5_ cluster is a necessary precondition for water oxidation [21,72]. The mechanism of photosynthetic water oxidation based on calcium has been discussed in detail in other reviews [73,74]. OEC splits water into oxygen molecules, protons, and electrons. Subsequently, electrons are transported to NADP^+^, generating NADPH via linear electron flow (LEF); LEF probably generates ATP and NADPH for the Calvin cycle [75]. In this process, photosystem 1 (PS1) can transfer electrons of ferredoxin (FD) to NADP^+^ and form NADPH through ferredoxin NADP^+^ oxidoreductase (FNR) [25].

However, if FD does not transfer electrons to NADP^+^ but passes through plastoquinone (PQ) to PS1 again, then this mode of electron transfer is called cyclic electron flow (CEF). PS1 participates in both LEF and CEF electron transfer, which plays an important role in the formation of CEF in electron transfer. PS1 is composed of multiple subunits (e.g., PsaA, -N, and -H) [76]. PsaN regulates photosynthetic electron flow through Ca^2+^-dependent phosphorylation [24], and it may also be related to electron transport from plastocyanin (PC) to PS1 [24]. PsaL and PsaA may also be associated with calcium [25]. FD is involved in the electron transport of PS1 [77] and has high affinity with Ca^2+^ in its reduced state [27]. In addition, FD can interact with PsaD, PsaE, PsaC, and PsaH [26].

In microalgae and vascular plants, CEF containing two parts: proton gradient regulation 5 (PGR5)/PGR5-like photosynthetic phenotype1 (PGRL1)-dependent pathway and NADPH dehydrogenase (NDH)-related pathway [7]. Munekage et al. [78] confirmed that PGR5 protein is an essential part of *A. thaliana* CEF components, and PGRL1 protein is found in *Rhine chlamydomonas* [79]. The CEF super complex is isolated from *R. chlamydomonas* containing PS1-light-harvesting (PS1-LHCI), cytochrome b_6_f complex (Cytbf), and FNR and PGRL1, and CAS is confirmed to be a part of the compounds [7]. CAS can interact with PGRL1 in vitro [7], and this interaction has a significant impact on CEF when CAS is downregulated in the Ca^2+^-dependent pathway [75]. Moreover, PGRL1 and CAS from *Chlamydomonas rheinensis* can interact with homologous proteins from *A. thaliana*, indicating that the interaction mode between PGRL1 and CAS is conservative [80].

The NDH complex has been found in plants and cyanobacteria [81], and NDH can regulate the balance of the ATP/NADPH ratio and prevent over-reduction in electron flow [81]. Peltier et al. [82] reported that the external variable sequence of type 2 of NADPH dehydrogenase (NDH-2) in plants contains an EF-hand motif that binds to Ca^2+^. NAD(P)H-dependent PQ reduction activity was found in the thylakoid membranes of potato and spinach [82]. In the chloroplast of *A. thaliana*, NADPH-dependent PQ reduction through the NDH-1 complex is strictly dependent on the presence of FD [82]. The NDH complex is activated by phosphorylating the NDH-F subunit and affects the dynamic levels of redox state of PQ [7]. In vitro experiments showed that the purified chloroplast protein kinase phosphorylates NDH-F subunits and is regulated by H_2_O_2_ and Ca^2+^ [83]. Other signaling molecules involved in linear and cyclic electron flow can be found in the review [7].

## 4. Ca^2+^ Involved in Regulating Photosynthetic Enzyme Activity of Carbon Assimilation

The Calvin cycle is the main pathway of carbon assimilation, and it occurs on the stroma of the chloroplast. Sedoheptulose-1,7-bisphosphatase (SBPase) and fructose-l,6-bisphosphatase (FBPase) are the two key enzymes in the Calvin cycle [28,29], and their activities are regulated by Ca^2+^ [30]. The two types of FBPase are cytoplasm FBPase and chloroplast FBPase [84]. The decrease in activity of chloroplast FBPase and SBPase can reduce the chloroplast content and inhibit plant growth, and the absence of these two enzymes in higher plants may damage photosynthesis [7]. Even though Ca^2+^ can regulate carbon assimilation by mediating these two enzymes, high concentrations of exogenous Ca^2+^ can inhibit carbon assimilation [85]. This phenomenon may be related to different experimental conditions, but the related mechanism remains to be further studied. Transketolase (TKL) is another key enzyme of the Calvin cycle, which occurs in the chloroplast [86]. It is involved in the regeneration of various substances in the Calvin cycle, such as erythrose4-phosphate (E4P) and xylulose5-phosphate (X5P) [86]. TKL was found to be phosphorylated in the chloroplast extract and was speculated to be related to the Ca^2+^-dependent pathway [31]. Additionally, CP12, a nuclear-encoded chloroplast protein with high Ca^2+^ affinity [34], can regulate the Calvin cycle by mediating the formation of the PRK/GAPDH/CP12 complex, which consists of phosphoribulokinase (PRK), CP12, and glyceraldehyde-3-phosphate dehydrogenase (GAPDH) [87]. Simultaneously, this complex can be affected by the ratio of NADP(H)/NAD(H) [32] and thioredoxin (TRX) [33].

## 5. Ca^2+^ Is Involved in the Mechanisms of Regulating Photoprotection

Under drought and other environmental stresses, a large number of stomata in plants will be closed. Thus, the reduction in CO_2_ entering the stomata, which cannot meet the demand of photosynthesis, is called a stomatal limitation factor. Stomatal movement is related to Ca^2+^ as described above. Non-stomatal limitation factors induce the decrease in photosynthetic efficiency caused by damage of photosystems under moderate or severe stress. Photoinhibition is also divided into two aspects: photodamage and photoprotection [88]. The photosynthetic proteins are damaged under light stress in an event called photodamage, such as the net loss of D1 protein.

Photoprotection refers to the capacity of preventing damage to the photosystem under excess energy, including the consumption of excess light energy and the removal of reactive oxygen species. During photosynthesis, the captured light energy is mainly used by photochemical electron transfer, chlorophyll fluorescence emission, and heat dissipation [89]. Photochemistry electron transfer is associated with the synthesis of photosynthetic products, and chlorophyll fluorescence emission is only rarely part of light energy consumption. Thus, heat dissipation is an important way to consume excess light energy and prevent photodamage. The process of dissipating harmlessly excess excitation energy as heat is called NPQ.

Some reports showed NPQ is regulated by Ca^2+^ [90]. The recombination of the PS2 reaction center complex is a common possible mechanism of NPQ, which involves reversible inactivation of D1 protein and synthetic regeneration [35]. High contents of D1 protein have been detected in Ca^2+^-treated plants [10], and the turnover of protein components of photosynthetic reaction centers is regulated by CaM, an important component of Ca^2+^ signal transduction pathway [91]. Moreover, Ca^2+^-binding sites may exist on D1 protein [36]. PsbS, a nuclear encoded PS2 subunit protein, also plays a key role in NPQ [37], because Ca^2+^ can induce the aggregation of PsbS in vitro [38]. In addition, the synthesis of violaxanthin de-epoxidase (VDE) is affected by Ca^2+^, and CaM mediates the expression of the *VDE* gene in the presence of Ca^2+^ to improve the xanthophyll cycle [10].

Actually, ROS are not only signaling molecules to regulate stomatal movement, but their excessive accumulation under stress can cause damage to plant cells. The ROS scavenging system can help plant cells maintain the balance of the ROS content. ROS can inhibit D1 protein recombination [92]. Exogenous Ca^2+^ can activate ascorbate peroxidase (APX), catalase (CAT), and superoxide dismutase (SOD) during heat stress [91]. Thus, the Ca^2+^ signal transduction pathway is involved in regulating ROS balance and protecting the photosystem.

## 6. Ca^2+^ Is Involved in Chloroplast Movement

In photosynthetic cells, chloroplast moves to different positions depending on light conditions. In weak light, chloroplasts are arranged evenly to follow light to absorb energy. In strong light, they are parallel to light to avoid light damage. Both blue and red light are important for photosynthesis, and chloroplast movement induced by blue light mediated and the pathway can cause a rise in intracellular Ca^2+^ levels in *A. thaliana* [93]. When *phot1* and *phot2* phototropin-deficient mutants are treated with PLC inhibitor neomycin and u-73122, the intracellular Ca^2+^ peak in *phot1* mutants suggests an increase by blue light induction, whereas *phot2* mutants are not significantly affected; these results indicate that PLC may mediate the phosphoinositol pathway to participate in the intracellular Ca^2+^ rise induced by phototropin2 [93]. Further studies showed that only phototropin2 is involved in the intracellular Ca^2+^ rise under strong blue light; by contrast, phototropin1 and phototropin2 are both involved in the intracellular Ca^2+^ rise caused by phosphatidylinositol 3-phosphate (Pl3P) under weak blue light, thereby causing chloroplast aggregation or the avoidance reaction [94]. Chloroplast motion involves filamentous actin, which is composed of aggregates of globular actin monomers, and this process is mediated by the Ca^2+^–CaM-dependent pathway [95].

Red light stimulates photoreceptor protein phytochrome to regulate chloroplast movement [96], but this regulation might be Ca^2+^-independent [97]. Further study discovered that chloroplast movement has two kinds of motion systems: one is microtubule-based and the other is microfilament-based; blue light may regulate both the microtubule movement system and the microfilament movement system, whereas red light refers to the microtubule movement system [96]. Thus, Ca^2+^ may eventually regulate the recombination of microfilaments to mediate chloroplast recombination [98].

## 7. Other Ca^2+^-Related Chloroplast Proteins 

As a common Ca^2+^-associated chloroplast CAS is associated with stomatal closure and electron chains (mentioned above), and it also participates in the chloroplast-mediated regulation of algae CO_2_ concentration [99,100]. Inorganic carbon in the atmosphere exists as CO_2_ for the photosynthesis of terrestrial plants. However, the content of CO_2_ in water is very small, and inorganic carbon mostly exists in the form of H_2_CO_3_. To adapt to the low concentration of CO_2_ in water, algae form a unique mechanism to rapidly absorb inorganic carbon from the external environment and convert it into CO_2_ in cells for photosynthesis. This mechanism is called CO_2_ concentration mechanism (CCM). This mechanism is involved in multiple proteins, such as the high-light activated 3 (HLA3) and low-CO_2_ (LC)-inducible protein A (LCIA), which are both involved in H_2_CO_3_ transport. These two proteins act together in H_2_CO_3_ transportation by transporting extracellular H_2_CO_3_ to the inorganic carbon pool of chloroplast stroma [100], and CAS is mainly used to regulate the expression of nuclear-encoded CO_2_-limiting-inducible genes, including *HLA3* and *LCIA* [100]. Recent research showed that CAS regulates CCM through the Ca^2+^-dependent pathway, which is not directly regulated by Ca^2+^ concentration but acts on the upstream of Ca^2+^ signal to regulate CCM [99].

Thylakoid-associated kinase (STN8) is an important photosynthetic protein in chloroplasts. STN8 dysfunction affects the phosphorylation of thylakoid membrane proteins and the expression of photosynthetic proteins encoded by nucleosomes and plastids [101]. STN8 is involved in the phosphorylation of PS2 proteins, including threonine phosphorylation at the N-terminal of D1, D2, and CP43 protein, and Thr-4 of PsbH [39]. Thr-4 is phosphorylated by STN8 only when Thr-2 is phosphorylated by other kinases [39]. The degradation rate of D1 protein in *Arabidopsis* mutants with STN8 kinase deletion is slower than that of wild type at high light [102], so the phosphorylation of D1 protein mediated by STN8 is involved in regulating PS2 repair mechanism in the case of photoinhibition [103]. CAS is also the phosphorylated substrate of STN8 [7]. Vainonen et al. [15] found that the phosphorylated level of CAS significantly increases under high-light stress. Phosphorylated CAS is highly likely to participate in signal transduction to respond to environmental stress, so CAS and STN8 may participate in part of the Ca^2+^-dependent signaling pathway.

Tic is a nuclear coding input chloroplast protein [40]. It cooperates with chloroplast outer envelope protein complexes (Toc) to pass cytoplasmic material through the chloroplast’s bilayer membrane. Tic is composed of several subunits, such as Tic110, Tic40, and Tic32 [104]. Tic110 can interact with Tic32, and they are both regulated by Ca^2+^ [41]. The C terminal of Tic32 has a CaM-binding domain, and the N-terminal has an NADP (H)-binding site, which might affect the photosynthetic pathway by regulating the amount of NADP (H) [104].

Guanosine 5’-triphosphate (or 5’-diphosphate) 3’-diphosphate [(p)ppGpp] is an important regulator in chloroplast function [42]. Research shows that (p)ppGpp levels affect photosynthetic capacity and chloroplast development in *A. thaliana* [105]. Genome analysis of *A. thaliana* found that four kind of RelA/SpoT homologue (RSH) enzymes from three families RSH1,2,3 can maintain the balance of (p)ppGpp in plants. RSH1 is mainly used as (p)ppGpp hydrolase, whereas RSH2 and RSH3 are mainly used as (p)ppGpp synthase [106]. Chloroplast localization proteins encoded by the *RSH* gene in rice were found to contain Ca^2+^ domains similar to EF-hand motif [42]. Thus, (p)ppGpp may be involved in regulating chloroplast function through Ca^2+^ signals. In *A. thaliana* RSH3-overexpressing lines, the accumulation of (p)ppGpp can rapidly reduce the number of chloroplasts coding rRNA and proteins, including PS2 supercomplex and other chloroplast complexes [107]. This result indicated that (p)ppGpp can regulate the expression of chloroplast genes by reducing the level of chloroplast transcription.

Chloroplast chaperones play an important role in cell chloroplast protein folding. The 60 kDa chloroplast chaperonin (ch-CPN60) and the 10 kDa chloroplast co-chaperonin (ch-CPN10) have been widely studied [43,108]. In previous studies, Yang and Poovaiah. [43] used ^35^S-labeled CaM experiments to find that CaM can bind to the C-terminal of *Arabidopsis* ch-CPN10. They also used Ca^2+^ chelating agent EGTA for a comparative experiment; their results demonstrated that CaM can only bind to *Arabidopsis* ch-CPN10 when Ca^2+^ is involved [43]. Moreover, *Arabidopsis* ch-CPN10 exhibits low similarity to the C-terminal of CPN10 in bacteria and mitochondria, so CaM may not bind to these proteins [43].

Many other chloroplast proteins are also regulated by Ca^2+^, such as SAMTL, which is located on the inner membrane of the chloroplast envelope and contains the EF-hand structure of Ca^2+^-binding sites [8]. Chloroplasts have a high demand for s-adenosylmethionine as a methyl donor for the synthesis of various substances, whereas SAMTL can transport s-adenosylmethionine into chloroplasts (8). ACA1 is a Ca^2+^ ATPase that resides in the inner envelope of *A. thaliana* chloroplast [44]. ATPase family gene 1-like protein 1 (AFG1L1) is a CaM-binding protein within the chloroplast [45]. These proteins are rarely investigated, so further details are not introduced here.

## 8. Conclusions

Calcium plays an important role in multiple photosynthetic pathways. It can affect gas exchange related to photosynthesis by regulating stomatal movement. Several photosynthetic proteins are regulated directly or indirectly by calcium. In addition to the proteins mentioned in this paper, a variety of Ca^2+^-related proteins located in the chloroplast outer membrane may directly link the cytoplasmic signal with the chloroplast signal. They are involved in multiple pathways responding to environmental stimulus (e.g., salt stress response and pathogen-associated molecular patterns) and regulate photosynthesis. The complex signal network related to Ca^2+^ needs further systematic research.

## Figures and Tables

**Figure 1 ijms-20-01353-f001:**
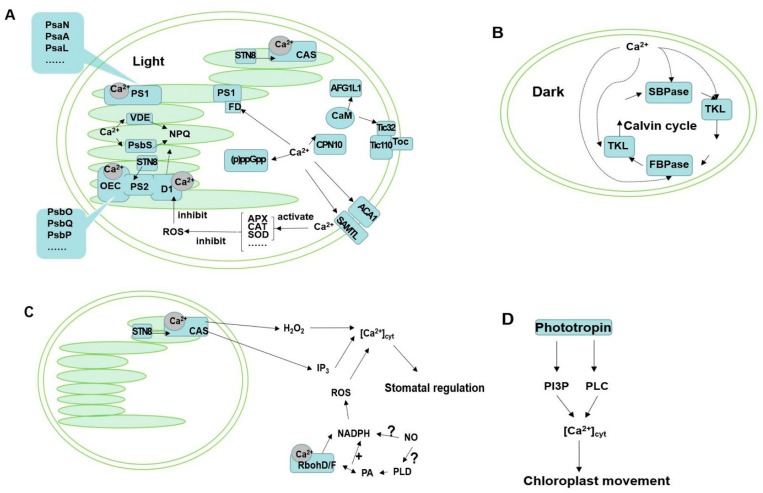
Photosynthesis-related pathways regulated by Ca^2+^. (**A**) Under light conditions, the proteins shown in this figure are all regulated by Ca^2+^, in which CAS is related to stomatal movement, photosynthetic electron flow, and CCM. PS1 subunits include PsaN, PsaA, PsaL, and PsaD. FD can interact with PS1 subunits to participate in electron transfer. PS2 is composed of OEC and D1 protein. OEC subunits include PsbO, PsbP, PsbQ, etc. OEC is involved in photosynthetic water oxidation. D1 protein, PsbS, and VDE are involved in energy dissipation. CPN10, Tic32, Tic110, ACA1, (p)ppGpp, and AFGLL1 play different roles in chloroplasts. (**B**) SBPase, TKL, and FBPase are the key enzymes related to Ca^2+^ in the Calvin cycle. (**C**) Stomatal movement is regulated by Ca^2+^. (**D**) Ca^2+^ is involved in chloroplast movement. Abbreviations: ACA1, *Arabidopsis thaliana* Ca^2+^-ATPase; AFG1L1, ATPase family gene 1-like protein 1; APX, ascorbate peroxidase; CaM, calmodulin; CAS, Ca^2+^ sensor; CAT, catalase; CCM, CO_2_ concentration mechanism; Ch-CPN10, 10 kDa chloroplast co-chaperonin; CP12, 12 kDa chloroplast protein; FBPase,fructose-l,6-bisphosphatase; FD, ferredoxin; IP_3_, the inositol 1,4,5-trisphosphate OEC, oxygen-evolving complex; PA, phosphatidic acid; Pl3P, phosphatidylinositol 3-phosphate; PLD, phospholipase D; PLC, phospholipase C; (p)ppGpp, Guanosine 5’ triphosphate (or 5’-diphosphate) 3’-diphosphate; PS1, photosystem 1; PS2, photosystem 2; ROS, reactive oxygen species; SAMTL, s-adenosylmethionine transporter-like; SBPase, sedoheptulose-1,7-bisphosphatase; SOD, superoxide dismutase; STN8, Thylakoid-associated kinases; Tic110, the subunit of chloroplast inner envelope protein complex; Tic32, the subunit of chloroplast inner envelope protein complex; TKL, Transketolase; Toc, chloroplast outer envelope protein complexes; VDE, violaxanthin de-epoxidase.

**Table 1 ijms-20-01353-t001:** Photosynthesis-related proteins associated with calcium.

Protein	Function Related to Photosynthesis	References	Function Related to Calcium	References
CAS	Stomatal regulation; photosynthetic electron flow; Regulate CCM	[12,13]	Ca^2+^-binding	[12,13,14,15]
RbohD/F	Stomatal regulation	[16,17]	Ca^2+^-binding	[18]
PsbO	The OEC subunit protein;	[19]	The Mn_4_CaO_5_ cluster as co-factor	[20,21]
PsbQ/PsbP	The OEC subunit protein;	[19,22]	The Cl^-^ and Ca^2+^ as essential co-factors	[19,23]
PsaN	Regulate photosynthetic electron flow	[24]	Regulated by Ca^2+^/CaM	[24]
PsaA/PsaL	The PS1 subunit proteins;	[25]	Possibly a Ca^2+^ coordinate the two proteins	[25]
FD	Electron transport of PS 1	[26]	High affinity with Ca^2+^	[27]
FBPase/SBPase	The Calvin cycle key enzymes	[28,29]	Regulated by Ca^2+^	[30]
TKL	The Calvin cycle key enzymes	[31]	Ca^2+^-dependent phosphorylation	[31]
CP12	Regulate the Calvin cycle	[32,33]	Ca^2+^-binding	[34]
D1 protein	Regulate NPQ	[35]	Ca^2+^-binding	[36]
PsbS	Regulate NPQ	[37]	Regulated by Ca^2+^	[38]
VDE	Regulate xanthophyll cycle	[10]	Regulated by Ca^2+^ and CaM	[10]
STN8	Phosphorylate thylakoid membrane proteins	[39]	Interaction with CAS	[7]
Tic110	Chloroplast inner envelope protein	[40]	Regulated by Ca^2+^	[41]
Tic32	Chloroplast inner envelope protein	[40]	Regulated by Ca^2+^	[41]
(p)ppGpp	The regulator in chloroplast function	[42]	Ca^2+^-binding	[42]
ch-CPN10	Assist chloroplast protein folding	[43]	CaM-binding	[43]
SAMTL	Chloroplast inner envelope protein	[8]	Regulated by Ca^2+^	[8]
ACA1	Chloroplast inner envelope protein	[44]	Ca^2+^ ATPase	[44]
AFG1L1	Chloroplast protein	[45]	CaM-binding	[45]

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
