# Peer review of "The Significance of Calcium in Photosynthesis"

_ijms, 2019, doi:10.3390/ijms20061353_

Reviewer 1 Report

An interesting review about an important subject such as the role of calcium in the regulation of the photosynthesis. The content is well focused and the treatment of the different parts of the review is adequate. The bibliography covers the state-of-the-art of the field.
The redaction of the review must be improved and the english style should be corrected throughout the text. Figure 1 should be modified as it looks like authors are talking about two different organelles and not of two processes taking place in the chloroplast. Full name of the different genes and proteins should be used the first time they are cited, otherwise it became difficult to follow the respective explanations

Author Response

Response to Reviewer 1 Comments

Thank you for your questions and suggestions.

Point 1The redaction of the review must be improved and the English style should be corrected throughout the text.

Response 1We have used a company that provides language-editing and copyediting services (http://essaystar.com/), and our manuscript has been revised very closely for mistakes and grammatical errors. 

Point 2Figure 1 should be modified as it looks like authors are talking about two different organelles and not of two processes taking place in the chloroplast.

Response 2We have divided Figure 1 into four parts (A, B, C, and D) to distinguish each process clearly.

Point 3Full name of the different genes and proteins should be used the first time they are cited.

Response 3We have added the full name of genes and proteins at first mention and provided an abbreviation list in the revised manuscript.

If there are any questions, please do not hesitate to reach us.

Reviewer 2 Report

The manuscript “The Significance of Calcium Involved in Photosynthesis” by Wang et al. presents a review on the role of Calcium (Ca2+) in photosynthesis. A very deficient English grammar hindered a full and fair evaluation of the content of the manuscript but, besides the linguistic problem (which revision is mandatory), a few drawbacks are apparent:

- A review on the same topic was made recently (Hochmal et al. 2015, cited in the manuscript); the authors must show what their manuscript add to this review; only circa 10 references in the manuscript list are dated after 2015; even though they include all the new references available on the subject, the detailed discussion of their content must be privileged over older references, already included in previous reviews.

- An abbreviation list (or at least the full designation of the acronyms on the first appearance on the manuscript) is necessary; many abbreviations wouldn’t be immediately understood by the average reader.

- Significant parts of the manuscript explain different physiological processes not directly related to Ca2+ function (e.g., lines 105-117 and 164-177); these parts should be eliminated or brought to the minimum required to understand Ca2+ action, since they don’t add value to the review and are written mostly in a text book style.

Author Response

Response to Reviewer 2 Comments

Thank you for your helpful comments and suggestions.

Point 1: A very deficient English grammar hindered a full and fair evaluation of the content of the manuscript.

Response 1: We have used the services of a language-editing and copyediting company (http://essaystar.com/) and proofread our manuscript carefully for mistakes and grammatical errors.

Point 2: A review on the same topic was made recently (Hochmal et al. 2015, cited in the manuscript); the authors must show what their manuscript add to this review.

Response 2: The review (Hochmal et al. 2015) mainly discussed the role of Ca2+ in the Calvin–Benson–Bassham (CBB) cycle, CAS protein, and linear and cyclic electron flow. We mainly discussed the mechanisms underlying the involvement of Ca2+ in stomatal movement, photosynthetic reaction, chloroplast movement, and in the mechanisms of regulating photoprotection.

Our manuscript discusses stomatal movement, photoprotection, and chloroplast movement. Two articles involve photosynthetic electron flow, so we cited them in our manuscript and added the new content from other references, include those dated after 2015. CAS protein is also discussed by Hochmal et al. (2015), and we added the role of Ca2+ in CO2 concentration mechanism. We did not discuss Calvin–Benson–Bassham in detail, but we cited the review of Hochmal et al. (2015) and added new content from other references dated after 2015. We also included content about Ca2+-dependent proteins, such as (p)ppGpp and ch-CPN10.

Point 3: Only circa 10 references in the manuscript list are dated after 2015; even though they include all the new references available on the subject, the detailed discussion of their content must be privileged over older references, already included in previous reviews.

Response 3: We searched the recent literature and found five available references dated after 2015. Thus, these articles have been included and cited in our manuscript. According to the revision, references were changed, i.e. some new references were added and some were deleted.

Point 4: An abbreviation list (or at least the full designation of the acronyms on the first appearance on the manuscript) is necessary.

Response 4: We have defined all acronyms at first mention in the paper and prepared an abbreviation list in the revised manuscript.

Point 5Significant parts of the manuscript explain different physiological processes not directly related to Ca2+ function (e.g., lines 105-117 and 164-177).

Response 5: We deleted the unnecessary sentences in lines 105–117. We opted to retain lines 164–177, because we want to discuss the involvement of Ca2+ in the mechanisms of regulating photoprotection. The concept of photoprotection is hard to distinguish from photoinhibition and photodamage. In lines 164–177, we describe stomatal movement, photochemical electron transfer, and removal of reactive oxygen species; these processes are related to calcium. Further clarification of these processes can facilitate readers’ understanding.

 If there are any questions, please do not hesitate to reach us.

Reviewer 3 Report

This is a fairly comprehensive mini-review highlighting the involvement of Calcium in different aspects of photosynthesis. There is a lot of room for improvement in sentence construction and English language usage.

Author Response

Response to Reviewer 3 Comments

Thank you for your helpful comments and suggestions.

Point 1: There is a lot of room for improvement in sentence construction and English language usage.

Response 1: We have used a company that provides language editing and copyediting services (http://essaystar.com/), and our manuscript has been proofread very closely for mistakes and grammatical errors.

 If there are any questions, please do not hesitate to reach us.

Round  2

Reviewer 1 Report

The different points have been addressed by authors

Reviewer 2 Report

Line 20: abbreviations should not appear in the keywords.

Line 87: “cytoplasm” instead of “cytoplast”

Sentences should not start with symbols and abbreviations.